# Failure Mechanism of the Fire Control Computer CPU Board inside the Tank under Transient Shock: Finite Element Simulations and Experimental Studies

**DOI:** 10.3390/ma15145070

**Published:** 2022-07-21

**Authors:** Xiangrong Li, Guohui Wang, Yongkang Chen, Bo Zhao, Jianguang Xiao

**Affiliations:** 1College of Mechatronic Engineering, North University of China, Taiyuan 030051, China; lxr118@163.com (X.L.); xiaojg@nuc.edu.cn (J.X.); 2Department of Arms and Control, Academy of Army Armored Forces, Beijing 100072, China; guohui305@126.com; 3Unit 32612 of the PLA, Guangyuan 628000, China; zhaobo8855@126.com

**Keywords:** dynamic analysis, modal analysis, harmonic response analysis, electronic component, fault mechanism

## Abstract

The electronic components inside a main battle tank (MBT) are the key components for the tank to exert its combat effectiveness. However, breakdown of the inner electronic components can easily occur inside the MBT due to the strong transient shock and large vibration during artillery fire. As a typical key electronic component inside an MBT, the fault mechanism and fault patterns of the CPU board of the fire control computer (FCC) are discussed through numerical simulation and experimental research. An explicit nonlinear dynamic analysis is performed to study the vibration features and fault mechanism under instantaneous shock load. By using finite element modal analysis, the first six nature frequencies of the CPU board are calculated. Meanwhile, curves of stress–frequency and strain–frequency of the CPU board under different harmonic loads are obtained, which are applied to further identify the peak response of the structure. Validation of the finite element model and simulation results are performed by comparing those obtained from the modal with experiments. Based on the dynamic simulation and experimental analysis, fault patterns of CPU board are discussed, and some optimization suggestions were proposed. The results shown in this work can provide a potential technical basis and reference for the optimization design of the electronic components that are commonly used in the modern weapon equipment and wartime support.

## 1. Introduction

With the extensive application of the high-performance electronic devices and control technologies, the automation and power of modern tanks have been greatly improved. However, the strong instantaneous shock and vibration during artillery firing can easily cause faults in electronic devices. The fault patterns are usually complicated, which involves mechanical deformation, contacts dislocation, sudden circuit interruption, devices failures, and so on. This makes it very difficult to maintain and support the main battle tank (MBT). Thus, understanding the fault mechanisms and fault patterns of these key components during tank artillery firing are essential to resolve these problems.

Dynamics studies on electronic devices under shock and vibration were started in the 1950s. Dave [1] systematically presents the vibration characteristics of all kinds of electronic components, and is still an important reference in the field related to shock and vibration of electronic devices [2,3,4]. During the 1980s, the finite element method (FEM) was introduced, and became the primary method to carry out vibration analysis of electronic devices. Pitarresi et al. [5,6] proposed and further improved the one region equivalent method, which effectively increased the computation accuracy and speed of calculating the natural frequencies and vibration modes of printed circuit boards (PCBs), and they also proposed five ways to set up the PCB finite element model. Yang [7] carried out research on the adaptability design in mechanical environment and dynamic reliability of space computer, and studied the structural dynamic reliability of device pins with random vibration excitation. Liu [8] proposed the methods of parameterized substructure modeling and PCB equivalent density modeling, which can greatly improve the simulation efficiency. Jeon et al. [9] built an isotropic elastoplastic FEM model for cell phone circuit boards and carried out both simulations and experiments with drop impact. Shtennikov and Budai [10] analyzed the circuit board welding points failure under vibration and proposed an effective improvement method. To enhance the reliability of electronic devices, Chen [11] investigated model transformation, boundary conditions and meshing on PCB modeling techniques in detail. Li [12] established a self-propelled artillery dynamics model to study the vibration characteristics and failure mechanism of the FCC under the launch shock and proposed the anti-vibration and cushion measures. Ding [13] carried out statistical analysis and failure cause analysis of impact failure cases of electronic equipment in aerospace equipment, and focused on the impact of important parameters such as peak value and frequency of impact response spectrum on electronic equipment failure. Xu [14] used impact dynamic response analysis to construct the impact damage boundary of plug-in components. The results showed that when the dominant frequency of the shock environment is higher than the first-order natural frequency of the SMA connector, its impact damage boundary is the relative displacement response asymptote of the shock environment. Xiang [15] explored a design method combining theoretical analysis and software simulation based on the mechanical environment stress design simulation, and performed a comprehensive simulation analysis and evaluation of the equipment’s structural strength.

In addition to the finite element method, the statistical energy method and transient statistical energy method are also widely used in impact response analysis [3,4]. These two methods do not require fine mesh division to be performed for the system, so there is no problem whereby the finite element method needs to subdivide the mesh when solving high-frequency problems, and can better deal with the high-frequency impact response of complex systems. However, these two methods are based on the statistical average of energy, so it is difficult to obtain the impact response at a specific location, and these two methods may fail when the modal density of the structure is small.

The shock from tank artillery firing is extremely strong and instantaneous, which can result in serious damage to internal electronic components inside the MBT. However, research on this topic has been rarely reported. Based on dynamics theory and FEM simulations, the computational analysis of faults mechanism of the CPU board in FCC (which is one of the important electronic components in the MBT) were performed. The results in this paper can be used as a helpful guide for the follow-up research on other electronic devices under strong and instantaneous shock during MBT artillery firing.

## 2. Basic Theory of Dynamics Analysis

Dynamics analysis is one of the important methods for structural analysis under instantaneous/dynamic load, and includes modal analysis, harmonic response analysis and transient dynamics analysis. Modal analysis is usually used to identify frequencies and vibration types of a structure. Harmonic response analysis is used to identify the response of the structure to steady-state harmonic load. Meanwhile, the transient dynamics analysis is used to identify the response of structure to load changing with the time [16].

The dynamics equilibrium equation is:*M**ü* + *I* − *F* = 0(1)
where *M* represents mass, *ü* represents acceleration, *I* represents internal forces which is determined by structure deformation and damping, and *F* represents external forces.

### 2.1. Modal Analysis

The mode is the inherent vibration characteristic of a mechanical structure. Each mode has a corresponding natural frequency, damping ratio and modal vibration type.

The mode can be calculated by
(2)$$\left(\left[K\right]-\omega_{i}^{2}\left[M\right]\left\{\phi_{i}\right\}\right)=0$$
where *K* represents stiffness, *M* represents mass, *ϕ_i_* represents the mode at stage *i*, *ω* represents vibration frequency of the mode at stage *i*. Please note that the modal analysis is the basis of harmonic response analysis and transient dynamics analysis.

### 2.2. Harmonic Response Analysis

Harmonic response analysis is used to identify the steady-state response of a linear object under simple harmonic loads. By calculating modal values for different frequencies and performing modal superposition method, the peak value can be obtained, and then the stress corresponding to the frequency can be analyzed.

The motion equation of harmonic response analysis is:(3)$$\left(-\omega^{2}\left[M\right]+i\omega \left[C\right]+\left[K\right]\right)\left(\left\{\phi_{1}\right\}+i\left\{\phi_{2}\right\}\right)=\left(\left\{F_{1}\right\}+i\left\{F_{2}\right\}\right)$$
where *C* represents damping, *F* represents loads, and *ω* represents vibration frequency.

The harmonic response analysis can be used to ensure that the research objects can withstand a variety of sinusoidal load with different frequencies. In addition, it can be used to obtain the resonant response which is either intended to avoid or make it happen, depending on each specific case.

### 2.3. Transient Dynamics Analysis

Transient dynamics analysis is a method used to identify the dynamic response of the structure subject to arbitrary loads changing over time. The stress, strain, and displacement of the structure can be obtained, which are usually changing with the time under transient loads.

The explicit nonlinear dynamic calculation in this work was performed in the FEM framework using the ABAQUS code. The central differential method is used to implement explicit time integration of the equation of motion (EOM), and the dynamic condition of the next step is calculated using that on the previous step. It also shows a high speed and good convergence of computation [17,18]. The process of transient dynamic calculation using the explicit dynamic analysis method is shown in the following.

A.Node calculation

The dynamic equilibrium equation is:(4)$${\ddot{u}|}_{\left(t\right)}={\left(M\right)}^{-1}{\left(P-I\right)|}_{\left(t\right)}$$
where *t* represents time.

The explicit integration to time is:(5)$${\dot{u}|}_{\left(t+\frac{\Delta t}{2}\right)}={\dot{u}|}_{\left(t-\frac{\Delta t}{2}\right)}+\frac{{\Delta t|}_{\left(t+\Delta t\right)}+{\Delta t|}_{\left(t\right)}}{2}{\ddot{u}|}_{\left(t\right)}$$

B.Cell calculation

According to the strain rate, Δ calculates the strain increment d*ε* of one cell. Then, according to the constitutive relation, the stress can be calculated as
(6)$${\sigma |}_{\left(t+\Delta t\right)}=f(\sigma_{(t),}d\varepsilon )$$
where *σ* represents stress.

The internal forces of cell nodes are integrated as *I*(*t* + Δ*t*).

C.Set time *t* as *t* + Δ*t*, then go back to step A.D.When *t* is equal to or larger than the preinstalled time, the calculation will stop.

## 3. Fault Mechanism Simulation and Analysis of CPU Board

The host module of FCC is a very important electronic device for the modern MBT, and includes the CPU board, I/O board, A/D board, power board, control board, display board and master unit. In this paper, a CPU board with a high fault rate is chosen as a typical research object, which will be simulated using FEM. The CPU board mainly consists of one CPU, one memory, data bus, control bus, interruption system circuit and reset system circuit, which is considered to be the “traffic light” in the FCC. It determines what functions should be stopped or started in the next step. It outputs signals to complete the determination by receiving the interruption or starting signal from other components such as the laser power counter, the I/O board, the A/D board or the surface board buttons. In addition, the operation and self-test programs of fire control and reset circuit of computer are stored in the CPU.

### 3.1. Modeling of CPU Board

Generally, there are five kinds of methods for PCB finite element modeling, including the simple method, the overall mass equivalent method, the overall mass and stiffness equivalent method, the overall and important parts equivalent method, and the complete modeling method [19]. The first three methods ignore the influence of small components on the board. Although the presence or absence of these small components has tiny influence on the vibration types of PCB, it can actually affect the value of the natural frequencies. The complete modeling method can yield simulation results with greater accuracy, but some shortcomings such as increased complexity of modeling and much more time-consuming simulation usually accompany this method. Therefore, the overall and important parts equivalent method is used in the paper. To make the simulation results more reliable, all components that are bigger than 10 mm are modeled using FEM, while other smaller components are ignored, such as the electric resistance and welding joints.

The established three-dimensional (3D) finite element model of the CPU board is shown in Figure 1. The size of the CPU board is 176 × 139 × 2 mm. There are nineteen chips, each of which includes one DSP chip, two CPU chips, and some small electric resistance and welding joints. Since the pins are not considered, connectors are modeled using an approximated cuboid geometrical model, so the circuit board and connectors are connected by using the conditions of multi-points constraint (MPC). Each element node represents a pin. The material properties for different parts are listed in Table 1.

### 3.2. Results of Modal Analysis

According to the actual boundary conditions of the circuit board, the needle-type plug base and the locking devices on the left and right sides are completely fixed. The first six natural frequencies and vibrational modes obtained by the modal analysis are shown in Table 2 and Figure 2, respectively.

A discussion regarding these simulation results is presented below.

(1)The first-order natural frequency of the PCB calculated in this work is 463.41 Hz, while the first-order resonance oscillation frequency of the computer case is 956.44 Hz, as shown in the previous computational study [20]. Thus, the design meets the multiplier rule. The previous modal analysis showed that large deformation occurred in some locations on the computer case that are used to fix the circuit board; thus, it is not easy for the CPU board and the computer case to resonate. However, a gap between the PCB and the slot can easily appear, leading to vibration.(2)The second-order natural frequency of the PCB reaches up to 998.65 Hz, which is slightly higher than the first order of the computer case. This indicates that the second mode or higher-order modes are not easily activated. Therefore, the effect of the first mode vibration should be mainly focused on for the design of the structure.(3)As shown in Figure 2, it can be found that the largest strain is located in the fore-end area of the board at low mode, and the strain gradually decreases toward the rear end. The minimum value of strain is located in the area near the rear end and the locking devices. Therefore, when designing boards in the future, important and relatively large parts should be placed as far away as possible from the board front, and should be settled close to the fixed areas.

### 3.3. Results of the Experimental Modal Analysis

To verify the simulation models and the corresponding results, experimental modal analysis was performed to obtain the natural frequencies and modal types of the CPU board. The principle of the modal experiment is to collect the excitation input and its corresponding output data through experimental measurement, and thus to obtain the modal parameters of the structure by fitting the experiment data points into the theoretical model. The flowchart of the modal experiment system is shown in Figure 3.

To obtain more accurate data, the original circuit board is fixed on the experiment platform according to the practical situation. Meshing of the board is shown in Figure 4.

A transient excitation method with a single point pulse hammer was used to motivate each grid node and obtain the output signal of the vibration response. For example, the output signal of node C21 is shown in Figure 5. Then, the modal types and natural frequencies were calculated by parameter identification for the single-reference-point frequency domain [21]. The comparison between the simulation and experiment results is shown in Table 3.

The coherence value of excitation and response signal is taken in the interval of (0.1). It is one of the important indicators of test quality and is a comprehensive evaluation parameter for the nonlinear impact of the structure, excitation force, noise pollution and frequency resolution. Generally, the coherence function should be greater than 0.8. By performing the orthogonality check for the coherence and vibration types of the response signal, it was found that the coherence value was 0.92 and the orthogonality value was 0.11. Both of these values are in accordance with the standard (GJB 2706A-2008, Modal test method), thus verifying that the experiment was effective and reliable.

### 3.4. Harmonic Response Analysis

Based on the results of the modal analysis, harmonic response analysis was carried out by modal superposition to obtain the curves of stress vs. frequency and strain vs. frequency of the CPU board. Different harmonic loads with different frequencies were considered, and the peak value of the dynamic response was found.

To obtain the response peak of the locations at which faults occur more easily, such as the chip pins and welding points, ten reference points (RPs) were defined on the board, as shown in Figure 6. RP 1, 4 and 7 are welding points, and the others are chip pins. The ten nodes corresponding to the ten RPs are listed in Table 4.

Three vibration curves of RP2 (UX_2, UY_2, and UZ_2) are shown in Figure 7, and the harmonic responses in the vertical direction (the Y-direction) of all RPs are obtained. The range of the frequency is 0~2000 Hz. The results show that the smaller thickness and lower strength of the circuit board can lead to higher amplitude in the vertical direction, and all peak values appear at the location of natural frequencies. The maximum amplitude is found at the first mode. Thus, it is clear that the most serious deformation is perpendicular to the circuit board surface.

### 3.5. Transient Dynamics Analysis

The time history of the acceleration in the Y direction of the FCC experiencing the shock and vibration caused by MBT artillery firing is shown in Figure 8. The acceleration was converted into a force that was changing over time, and then applied to the PCB. The integral time step was set as 0.001 s, and transient dynamics analysis was carried out using the full simulation method [22,23]. The stress response curve of RP 2 is shown in Figure 9. By comparing the curves between Figure 8 and Figure 9, it can be found that the peak values of both curves appear almost simultaneously at the time of 0.48, 0.8 and 1.3 s. The stress of RP 2 reaches its highest value at 1.3 s, which is close to 25 MPa.

### 3.6. Suggestions for Design Optimization

According to the results of modal analysis, harmonic response analysis and transient dynamics analysis of the PCB, we were able to gain some insight into the dynamic properties and fault features of PCB of FCC under the assumed strong and transient shock during artillery firing.

The PCB and the computer case meet the multiplier rule at the first natural frequency, and they do not resonate easily. However, some locations on the case used to fix the circuit board deform seriously, which can easily result in a gap between the PCB and the slot, leading to vibration.

The locking devices are located at the positions on the rear end on both transverse sides of the board. In addition, there is only a connector in the fore-end as a stiffened support. The faults statistics show that the circuit board connector would become loose or pop under the shock [24]. Therefore, the fore-end of the board, especially both transverse sides, can be considered to be in a suspended state. On the basis of the dynamics simulation results shown above, it can be found that the stress and strain responses of the fore-end area are much stronger, and the deformation is much more severe. This is in good agreement with the actual measured fault statistics, which show that faults like cracks and punctures evidently appear more easily at the fore-end area.

In regard to the above fault features of PCB, some optimization suggestions are proposed. In the future design of this structure, it would be better to:(1)Increase the fixing points on the board and set locking devices on both sides of the fore-end area to make it no longer suspended.(2)Add high-strength tendons on the fore-end area to enhance the toughness of the PCB, and increase the thickness of the PCB or use a kind of higher elasticity and intensity material to enhance the bending resistance performance.(3)Place the components on the board closer to the locking devices or the tendons. The dynamic analysis shows that the closer the component is to the fix point, the weaker the vibration response of board becomes, and the less damage occurs to the chips and resistors.

Taking the first suggestion as a research object, verified simulation analysis was further implemented. As shown in Figure 10, each locking device on both sides was divided into three spatially separated parts placed evenly, with average intervals along the edge of the board. The results of the modal analysis are shown in Figure 11, and the first six natural frequencies are listed in Table 5.

From the vibration types of mode shown in Figure 11, it can be seen that for the vibration types of I, IV and V, the structure deforms longitudinally, while for vibration types of II, III and VI, it deforms transversally. Considering the first natural frequency and its corresponding mode as an example, the deformation trend after optimization decreased, while still meeting the multiplier rule. The deformation trend of the board displacement after optimization changes from transverse to longitudinal. The stress contours in Figure 11 show that the region with the most severe deformation is significantly reduced, and is mainly located at the center of the fore-end area. There is still some deformation at the center of the board. However, this can be ignored, since the value of displacement deformation is tiny. The analysis results indicate that the deformation of the board can be reduced effectively, and the reliability can be improved through designing a reasonable distribution of locking devices.

## 4. Conclusions

In summary, the dynamic characteristics and fault mechanism of the CPU board under strong and transient shock during MBT artillery firing were studied by utilizing FEM simulations and dynamic analysis. The conclusion remarks are summarized as follows,

(1)The first six natural frequencies and vibration modes were calculated in the FEM simulations and verified by experiments. The results of harmonic response analysis show that smaller thickness and lower strength of the PCB can result in larger response amplitude in the vertical direction. All peak values appear at the natural frequencies, and reach their maximum at the first frequency (463.41 Hz), and the most serious deformation appears in the direction perpendicular to the board surface.(2)The results of transient dynamics analysis show that peak stresses appear at the time of 0.48, 0.8 and 1.3 s and the maximum value is close to 25 MPa at 1.3 s. The first peak is directly resulted from firing shock, and the second and the third peak appear during aftereffect. Therefore, more attention should be paid to the aftereffect so as to avoid faults in electrical components.(3)The dynamic simulation shows that a gap between the CPU board and the slot can easily occur, leading to large vibrations throughout the entire structure. The fore-end area exhibits much stronger stress and strain responses with a larger deformation, which is consistent with the actual fault statistics [14].(4)On the basis of this research, some optimization suggestions are proposed. Increase the fixing points and set locking devices on both sides of the fore-end area so that it is no longer suspended. Add some high-strength tendons to enhance the PCB’s toughness. Increase the thickness or use a kind of higher elasticity and intensity material to enhance the bending resistance performance. Settle the components on the board closer to the locking devices or the tendons.(5)Due to technical and time constraints, this paper only analyzes the mechanical failure mechanism of the CPU board. However, the number of tank electronic control components is large, and the types are miscellaneous. More electronic control components, such as the program control box of the automatic loader and the night vision device of the gunner, should be discussed. At the same time, secondary fault modes such as short circuit, ablation and breakdown should be considered, and the transmission fault mechanism of the internal circuit of the electronic control components under the impact should be studied by electromechanical joint simulation.

## Figures and Tables

**Figure 1 materials-15-05070-f001:**
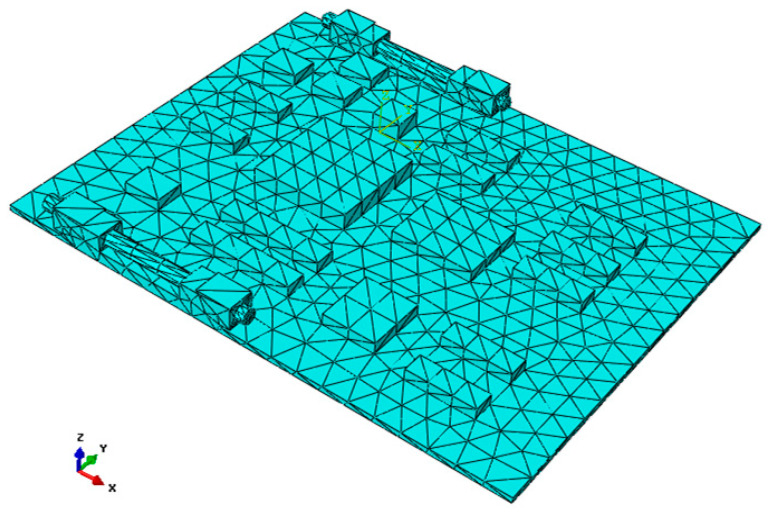
Finite element model of the CPU board.

**Figure 2 materials-15-05070-f002:**
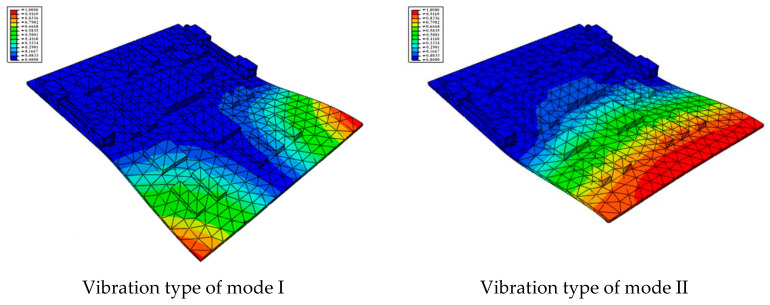
The first 6 modes vibration type based on stress of CPU finite element model.

**Figure 3 materials-15-05070-f003:**
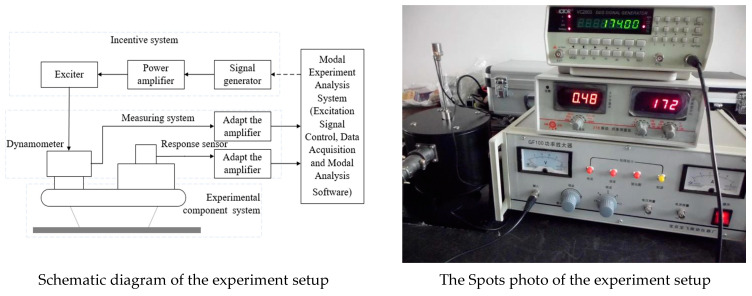
Flowchart of the mode experiment system.

**Figure 4 materials-15-05070-f004:**
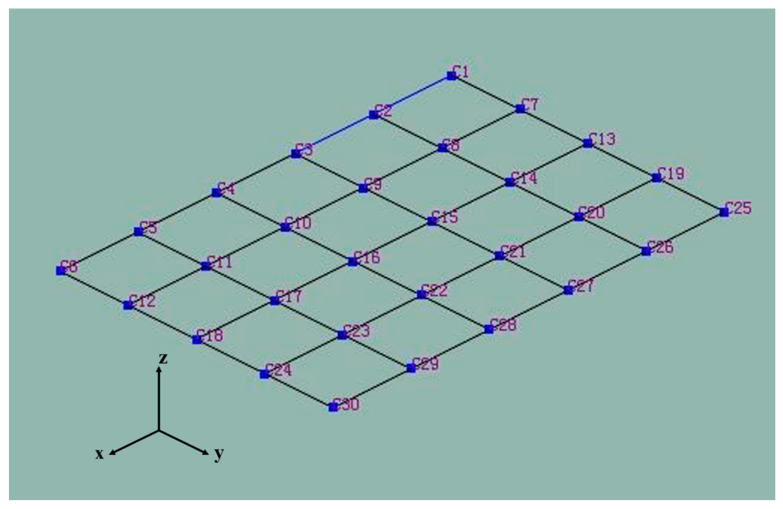
Meshing during the mode experiment.

**Figure 5 materials-15-05070-f005:**
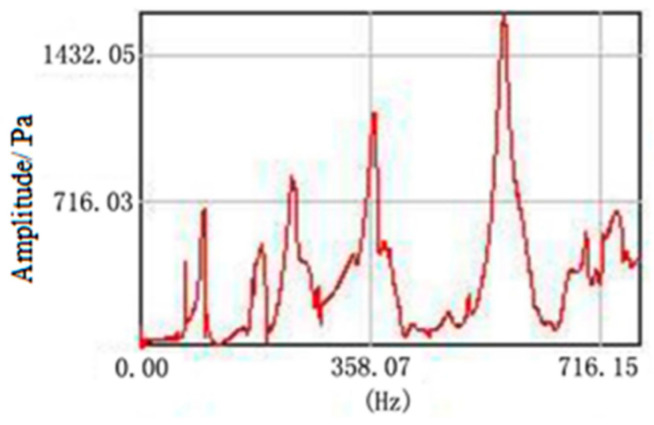
Response output of node C21.

**Figure 6 materials-15-05070-f006:**
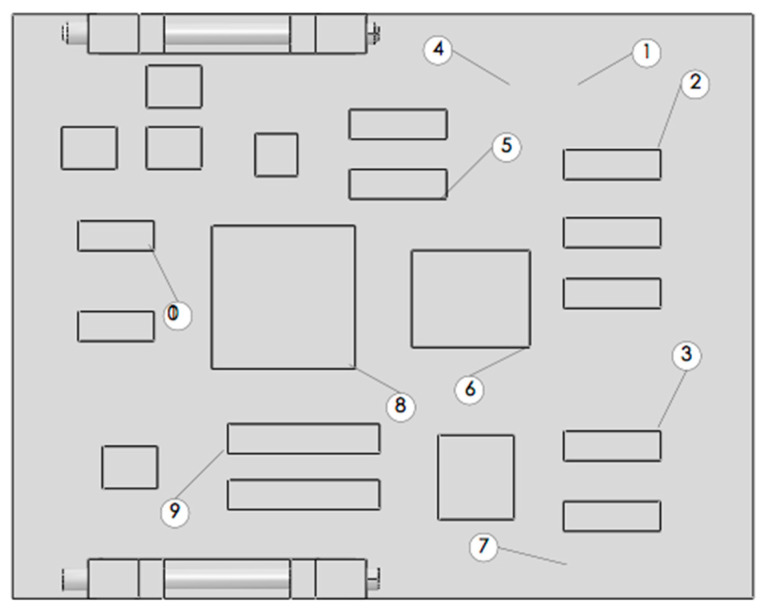
Ten reference points defined in the CPU board.

**Figure 7 materials-15-05070-f007:**
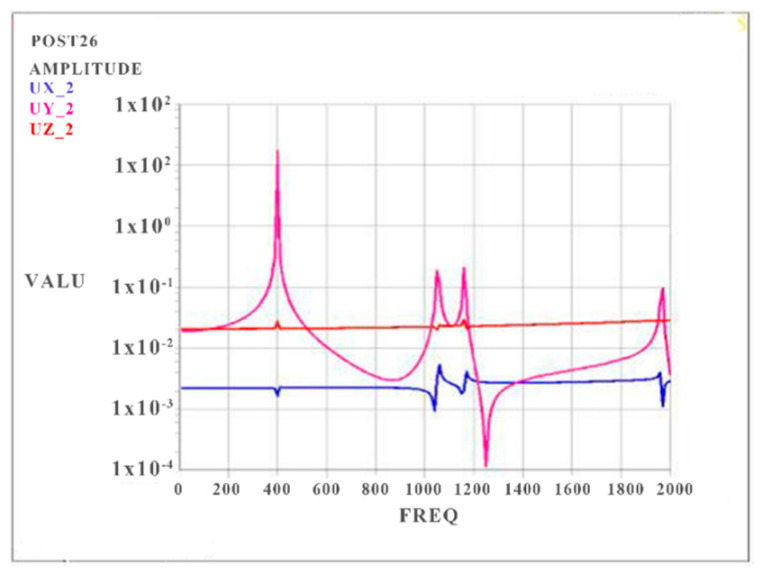
Harmonic response curve of RP2.

**Figure 8 materials-15-05070-f008:**
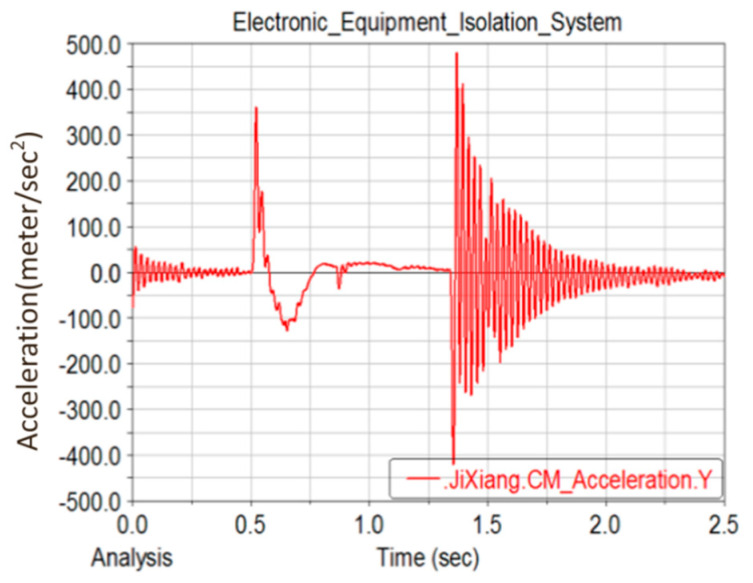
Acceleration in Y direction of FCC under shock and vibration caused by MBT artillery firing.

**Figure 9 materials-15-05070-f009:**
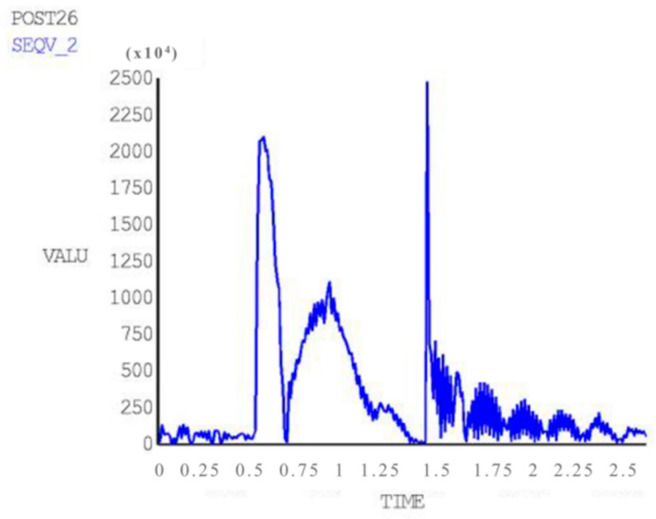
Stress response curve of RP 2.

**Figure 10 materials-15-05070-f010:**
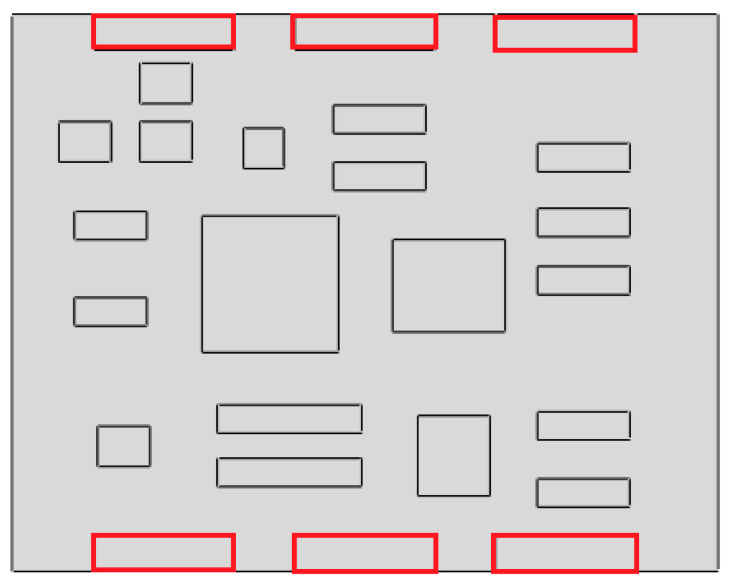
Circuit board locking device optimization.

**Figure 11 materials-15-05070-f011:**
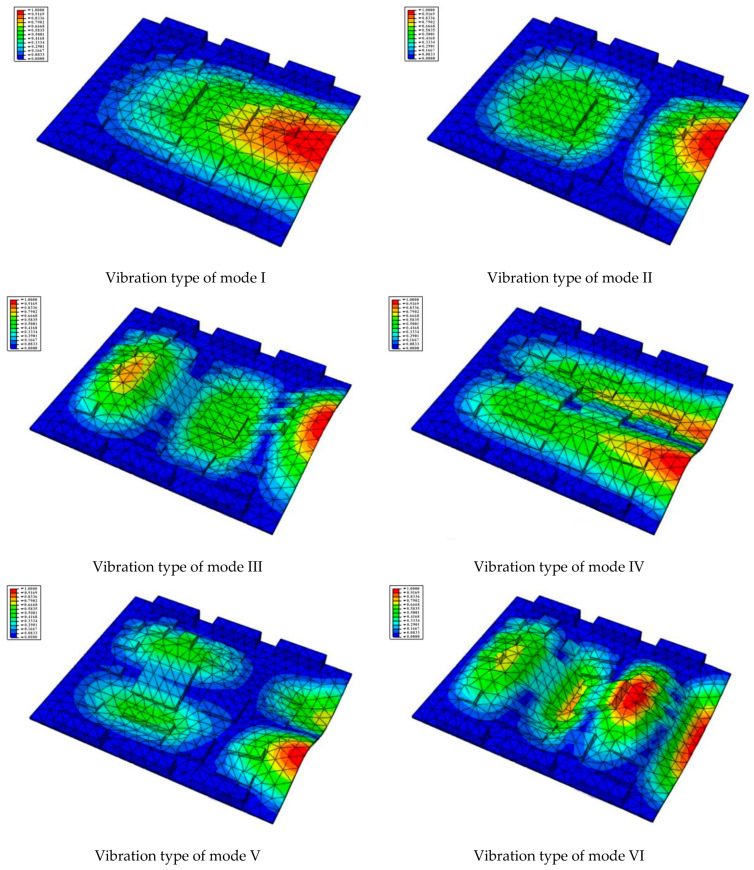
The first six vibration types of CPU board after optimization.

**Table 1 materials-15-05070-t001:** Material parameters of the CPU board model.

Parts	Density (kg/m^3^)	Elasticity Modulus (GPa)	Poisson’s Ratio
PCB	1800	11	0.28
The chip substrate	1350	18.62	0.18
Welding joints	7500	15	0.32

**Table 2 materials-15-05070-t002:** First six natural frequencies of CPU board.

Vibration type	I	II	III
Frequency/Hz	415.41	998.65	1325.7
Vibration type	IV	V	VI
Frequency/Hz	1935.7	2501.3	2814.1

**Table 3 materials-15-05070-t003:** Comparison between experiment and simulation of PCB mode analysis.

Frequency (Hz)	Simulation Mode	Frequency (Hz)	Experimental Mode	Relative Error
415.41	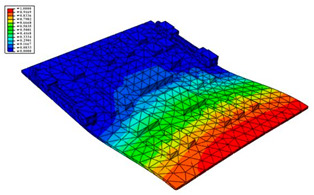	401.70	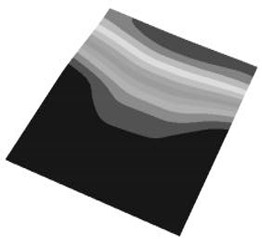	3.3%
998.65	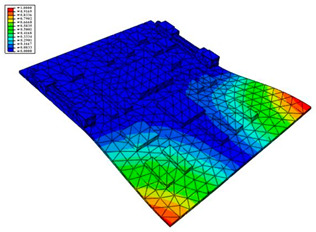	969.29	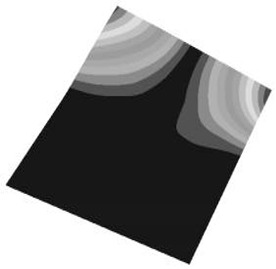	2.94%
1325.7	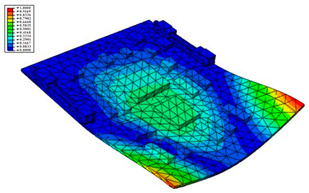	1296.93	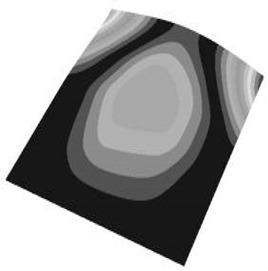	2.17%
1935.7	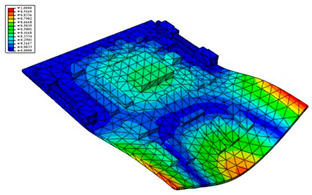	1808.33	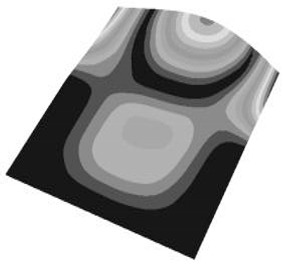	6.58%

**Table 4 materials-15-05070-t004:** Ten reference points corresponding to the nodes.

Reference Point	Node	Reference Point	Node
1	354	6	1403
2	733	7	2544
3	1014	8	4368
4	437	9	5629
5	1355	10	7323

**Table 5 materials-15-05070-t005:** The first six frequencies of the CPU board.

Vibration type	I	II	III
Frequency/Hz	306.40	768.26	1532.48
Vibration type	IV	V	VI
Frequency/Hz	1842.25	2611.13	3436.28

## Data Availability

The data presented in this study are available on request from the corresponding author. The data are not publicly available due to the privacy of the research object.

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
