# Peer review of "Failure Mechanism of the Fire Control Computer CPU Board inside the Tank under Transient Shock: Finite Element Simulations and Experimental Studies"

_materials, 2022, doi:10.3390/ma15145070_

Round 1

Reviewer 1 Report

Manuscript Number: materials-1771446

Full Title: Failure Mechanism of the Fire Control Computer CPU Board inside the Tank under Transient Shock : Finite Element Simulations and Experimental Studies

The authors investigated as a typical key electronic component inside  the main battle tank , the CPU board of the fire control computer (FCC) in this paper. The authors claim that the article's results shown in this work can provide a potential technical basis and reference for the optimization design of the electronic components that are commonly used in the modern weapon equipment and wartime support. My reviews and suggestions about their publications are listed;

1) The abstract should include the context or background information for your research; the general topic under study; the specific topic of your research; why is it important to address these questions; the significance or implications of your findings or arguments. It must also contain more numeric values. Please highlight your contribution. Reorganize the abstract to conclude:
(a) The overall purpose of the study and the research problems you investigated.
(b) The basic design of the study.
(c) Major findings or trends found as a result of the study.
(d) A brief summary of your interpretations and conclusions.

2)The introduction should be extended to the more detailed background that would be supported by some literatüre. The contributions of the article should be presented in more detail. Also, they need to be reconsidered. Some are not contributions. The introduction needs to clarify the (1) motivation, (2) challenges, (3) contribution, (4) objectives, and (5) significance/implication.

3) The literature presented by the authors is insufficient and outdated. Add more recent reference to enhance literature survey section. Discuss the state-of-art techniques with their merits and issues. The literature should be developed and, if possible, presented in papers published in 2020, 2021 and 2022. Literature Review section can be further strengthened by discussing various recent related article. Discuss the research gaps and relate how the proposed work has improved them. Literature Review section can be further strengthened by discussing various recent related article. Discuss the research gaps and relate how the proposed work has improved them.

4) You should submit more experimental study results for your work. Experimental studies are insufficient. You should also provide comparisons with similar studies.

5) What solution you propose to make the system more robust. What is your difference from similar studies?

6) Conclusion section should be extended. In other words, the author should discuss the future works related to proposed method and its drawbacks. Rewrite the conclusion with following comment:

(a) Highlight your analysis and reflect only the important points for the whole paper.

(b) Mention the implication in the last of this section. Please, carefully review the manuscript to resolve these issues.

(c) This section should be supported with numerical values.

7) There are some typos and grammar errors in the manuscript. The quality of the article should be increased.

Reviewer 2 Report

This is a well-written paper with an interesting topic. Here are my comments to the authors:

1- Past tense should be avoided in the abstract.

2- It is required to manage the Tables in a way that all of the rows are placed on a single page.

3- All of the references are rather old. The authors are required to search for new articles, review state of the art, and update their references.

4- The numbers in almost all of the figures are either too small or too pale to be readable.

5- All of the boxes in Fig. 3 must have a label to be explicitly defined.

Reviewer 3 Report

The manuscript presents an explicit nonlinear dynamic analysis to investigate the vibration features and fault mechanism of CPU board of MBT under instantaneous shock load. The corresponding natural frequencies and peak responses were determined using FE method. The faults pattern performance of the board is numerically analyzed and then compared with that of experiment.

General comment: There are some typos and grammatical errors throughout the manuscript. It is highly recommended to re-read and re-revise the manuscript in order to make it suitable for the readers.

 There are also several major concerns remaining related to the technical perspectives that need to be addressed, such as:

1.      It is recommended to add recent references that can confirm the following statement:

“Dave[1] presented systematically the vibration characteristics of all kinds of electronic components, which is still an important reference in the field related to shock and 38 vibration of electronic devices.”

2.      The introduction of the paper is not enough to confirm the contribution of the present work and must be substantially improved by further analysis of recent works and adding recent published articles about the topic.

3.      The parameters in all the equations should be defined.

4.      It is recommended to add the dimension of the CPU presented in Fig.1.

5.      The resolution of figure 2 is very low and needed to be improved.

6.      The flowchart presented in Fig. 3 is not easy to follow.

7.      The orthogonality value should be defined.

8.      The authors mentioned that “Both of them are in accordance with relevant standard, 200 which can verify that the experiment is effective and reliable.”. It is highly recommended to add some standards to confirm the statement.

9.      The picture of the experimental test setup should be added to the manuscript.

10.  The figure 7 and 8 can not be read.

11.  Figure 12 also cannot be followed.

It is important to note that all the comments must be addressed in the revised manuscript.

It is also very important to discuss about the similar FE and experimental works on the CPU and compare the current results with them.

Round 2

Reviewer 1 Report

I have reviewed the revised manuscript title "Failure Mechanism of the Fire Control Computer CPU Board inside the Tank under Transient Shock : Finite Element Simulations and Experimental Studies". After revising my initial comments and comparing the changes, done by the authors, with them. I found that the authors addressed and answered most of the comments efficiently. Overall, the revised manuscrip is well organized and carefully prepared. The response letter was elegant and satisfactory. I thank the authors for their kind responses. The authors have sufficiently address my all comments. So, I think it is appropriate to accept the revised article. The authors have addressed all the concerns and responded to the review comments. The manuscript can be published in this journal.

Author Response

Thank the reviewer for your careful guidance of this article. Your comments have effectively helped improve the level of this article and provided direction and guidance for our follow-up research.

Reviewer 3 Report

 I, as the reviewer, appreciated much the sincere response made by the authors to my comments.    Many readers should be interested in reading and understanding the presented results. Actually Figures 8,12, and Table 3 seems to have shown a little bit improvement but still cannot be read. The reviewer expected that the authors would make some improvement on this issue.

However, I am afraid that the authors do not seem to care about visibility of the presented figure. Therefore, the reviewer will not pursue this issue any more.

Author Response

On behalf of my co-authors, we are very grateful to you for your positive and constructive comments and suggestions on our manuscript entitled “Failure Mechanism of the Fire Control Computer CPU Board inside the Tank under Transient Shock : Finite Element Simulations and Experimental Studies” (ID: materials-1771446).

We have tried our best to revise the manuscript. Figures 8,12, and Table 3 are the clearest version we can provide. Thank you for your tolerance and understanding. We continue to polish and revise the article. Your comments have effectively helped improve the level of this article and provided direction and guidance for our follow-up research.

This manuscript is a resubmission of an earlier submission. The following is a list of the peer review reports and author responses from that submission.